# Ideas and Perspectives: Potentially Large but Highly Uncertain Carbon Dioxide Emissions Resulting from Peat Erosion

Thomas C. Parker<sup>1</sup>, Chris Evans<sup>2</sup>, Martin G. Evans<sup>3</sup>, Miriam Glendell<sup>1</sup>, Richard Grayson<sup>4</sup>, Joseph Holden<sup>4</sup>, Changjia Li<sup>5</sup>, Pengfei Li<sup>6</sup> & Rebekka R. E. Artz<sup>1</sup>

- <sup>5</sup> The James Hutton Institute, Craigiebuckler, Aberdeen, AB15 8QH, UK
  - <sup>2</sup>UK Centre for Ecology and Hydrology, Bangor, LL57 2UW, UK
  - <sup>3</sup>Faculty of Social Sciences and Health, Durham University, Durham, DH1 3LE, UK
  - <sup>4</sup>School of Geography, University of Leeds, Leeds, LS2 9JT, UK
  - <sup>5</sup>Faculty of Geographical Science, Beijing Normal University, Beijing 100875, China
- 10 <sup>6</sup>College of Geomatics, Xi'an University of Science and Technology, Xi'an, 70021, China

Correspondence to: Thomas C. Parker (Thomas.parker@hutton.ac.uk)

#### **Abstract**

Peatland erosion and resulting particulate organic carbon (POC) flux is an international problem that is causing loss of peat carbon to the atmosphere and contributing to climate change. Peatlands from around the world are eroding and losing carbon for a range of reasons, from overgrazing to climate change, and the POC is subsequently exposed to a diverse range of conditions, depending on the geographical context. The context, drivers of erosion and downstream environment will directly influence the rate at which POC is mineralised to CO<sub>2</sub> by microbial communities. Despite the potential large carbon losses from POC and subsequent CO<sub>2</sub> emissions, the mechanisms for emissions reporting at international and national level are undeveloped. Here we highlight the key limitations for understanding and quantifying emissions that result from peat erosion and discuss the research that is required to address these limitations. We particularly consider quantification of direct CO<sub>2</sub> emissions from bare peat and resedimentation and further turnover at different scales. By integrating biological and geomorphological process understanding we can work towards better quantification of peatland emissions and the emissions that can be avoided through peatland ecosystem restoration.

#### 25 1 Peatland Erosion and Potential Emissions

Peatlands have been under severe threat from various anthropogenic factors such as pollution, drainage and intensive utilisation for food and fibre production and many are now strong net sources of carbon to the atmosphere (Evans et al., 2021b). Peatlands are important sources of fluvial carbon including particulate organic carbon (POC), dissolved organic carbon (DOC) and dissolved gases (Billett et al., 2015; Rosset et al., 2022). Previous studies have suggested that the relative roles of these fluvial forms are typically ~15-40% of CO<sub>2</sub> equivalent net ecosystem exchange (Dinsmore et al., 2010; Roulet et al., 2007; Billett et al., 2010). However, POC flux is particularly high from peatlands where vegetation cover is partial (Evans et al., 2006) and in

these systems POC can contribute > 80 % of the fluvial flux (Pawson et al., 2008) while a lack of vegetation will also be associated with a reduced terrestrial C uptake across the peatland and potentially to enhanced direct losses to the atmosphere. Given such large potential contributions to C losses, it is critical that more studies acknowledge the POC pathway in the carbon budget. Previous studies have suggested that both DOC and POC are metabolised to CO<sub>2</sub> in the fluvial system to some degree, with current best estimates between 50 – 90% conversion for POC and 80 -100% for DOC (Evans et al., 2013). However, most studies focus on terrestrial gas fluxes or aquatic DOC fluxes. Hence, the various pathways for POC storage, transport or transformation to CO<sub>2</sub> are not well studied (Palmer et al., 2016). A large body of research has examined the geomorphological controls on peat erosion, especially from the perspective of sediment load (Reviewed in Li et al. (2018b)). However, as governments and landowners look to quantify and reduce greenhouse gas emissions from peatlands, focus should turn to quantifying those that result from peatland erosion.

The onset of peatland erosion can be traced back over a thousand years (Evans and Warburton, 2011). It is hypothesised that there is a 'threshold process' whereby the peat changes from a stable, intact state to an unstable, erosional state. Some propose that erosion is a natural termination after thousands of years of peat accumulation resulting in instability of the peat mass (Conway, 1954; Pearsall, 1956; Colhoun et al., 1965). Others argue that much of the erosion has resulted from anthropogenic pressures, including burning (Yallop et al., 2009), overgrazing (Wilson et al., 1993), artificial drainage installation (Worrall and Evans, 2009; Holden et al., 2007), and atmospheric pollution (Yeloff et al., 2006). Climate change can potentially enhance rates of erosion through more extreme weather events (e.g. Cotterill et al. (2021)). For example, drought can cause desiccation of the peat, and impact by heavy rain can cause 'wind splash' impacts and rapid overland flow, contributing to destabilisation and transport of the peat (Warburton, 2003). In contrast, reduced frost days and fewer freeze-thaw cycles could reduce erosion caused by needle ice (Li et al., 2018a). Interactions between climate change and erosion are complex and often ecosystem/biome or region specific. Nonetheless, erosion of peat is projected to change in the coming century with sediment yields projected in different regions decreasing or increasing by -1.27 to +21.63 t ha<sup>-1</sup> yr<sup>-1</sup> by 2800 (Li et al., 2017). It is therefore important to understand the links and feedbacks between climate, land use, peat erosion and CO<sub>2</sub> emissions.

55

In the past century, most of the peat erosion and post-erosion POC research has been conducted in the UK. However, peat erosion is a pressing or emerging problem for peatland systems around the world, with potential for massive carbon losses and climate feedbacks (Fig.1). Potential erosion hotspots are occurring in different environmental and management contexts around the world from drained forestry sites which may have relatively low areas of exposed peat (Marttila and Klove, 2010) to industrial extraction sites with almost complete bare peat cover (Campbell et al., 2002). At the extreme end of erosion, collapse of inland permafrost systems in the arctic and boreal regions (Swindles et al., 2015) can cause localised rapid erosion and movement of soil carbon via thaw slumps (Lamoureux et al., 2014; Pizano et al., 2014), with potential for high emissions as the mobilised carbon becomes available to decomposer organisms in freshwater environments (Li et al., 2024). In contrast, arctic permafrost coastal erosion and coastal-adjacent thaw slumps, which are occurring at an alarming rate in response to

rapid warming around the Arctic Ocean, are depositing carbon directly into the ocean (Lantuit and Pollard, 2008; Lantuit et al., 2012). Equally, In the tropics of Asia, coastal erosion of peatlands is causing large direct fluxes of peat to the ocean (Kagawa et al., 2024), but there are also examples of inland peat erosion in Asia which will generate POC that is primary processed in terrestrial systems (Wang et al., 2019).

Peat erosion is clearly progressing in a variety of contexts and at different rates, but in every case it will be exacerbated by climate change and associated extreme weather events (Zhao et al., 2024). This is why IPCC reporting of emissions needs to move towards a more nuanced understanding of POC turnover than the broad downstream POC-CO<sub>2</sub> conversion rate of 70% (based on UK examples (Ipcc, 2014)). Depending on the context, biome and global location, estimated emissions resulting from peat erosion could vary significantly from currently reported rates.

Figure 1: Bare peat in a. Scotland (Photo T. Parker), b. Lesotho (Photo C. Evans), c. Falkland Islands (Photo C. Evans), d. Siberia (©ESA; Photo Annett Bartsch) and e. Indonesia (Photo S. Smith).

The 2013 IPCC wetlands supplement (Ipcc, 2014) present a general calculation for a POC emissions factor (EF<sub>POC</sub>) for all peatlands and drained organic soils. This generic model, although primarily based on evidence from the UK, was designed for

any peatland soil that had suffered significant disturbance that lead to bare peat, including drainage, burning, peat extraction and conversion to arable land as follows (Ipcc, 2014):

$$EF_{POC} = POC_{FLUX BAREPEAT} \times PEAT_{BARE} \times Frac_{POC-CO_2}$$

Where: **POC**<sub>FLUX BAREPEAT</sub> is the POC flux per area of bare peat surface, **PEAT**<sub>BARE</sub> is the area of bare peat and **Frac**<sub>POC</sub>-CO<sub>2</sub> is the conversion of POC to CO<sub>2</sub> following export from the peatland.

Mapping of bare peat extent at high resolution is progressing (Macfarlane et al., 2024) but the underpinning data for estimating emissions associated with bare peat are highly uncertain (Evans et al., 2013) as the flux depends on specific fluvial mixing events in time and space (Palmer et al., 2016). We argue that this calculation has two major sources of uncertainty which are critical to resolve to confidently quantify emissions that arise from peat erosion. Firstly, the flux of POC from bare peat at the source is only one part of peat volume loss - quantification of the relative contribution of direct CO<sub>2</sub> loss, subsidence and erosion to surface retreat rates will give rise to better quantification POC loss via erosion. Eroded peat will potentially be processed and mineralised in multiple environments, from headwater streams, floodplains to rivers and the ocean (Evans et al., 2013; Zhou et al., 2021). Therefore, the second source of uncertainty is the fraction of eroded peat/POC that is converted to CO<sub>2</sub>. This can be addressed by considering the environments and organisms that interact with it while in transit over various timescales.

# 2 When peat is exposed to the atmosphere, how much carbon is lost as CO<sub>2</sub> compared to other mechanisms of peat volume loss?

#### 2.1 Erosion

85

Peat sediment can be destabilised by needle-ice production, wind-splash and desiccation and then transported offsite by rill and interrill water erosion and wind erosion. Bare peat in gullies or flats is under significant erosion pressure from these processes (Li et al. (2018b); Evans and Warburton (2008)). For the purposes of this article we do not consider wind erosion because although lateral transport may be significant (0.46-0.48 t sediment ha<sup>-1</sup> yr <sup>-1</sup> (Warburton, 2003)), in the absence of more information we assume the wind-eroded peat is retained within the erosion system. This is likely true in winter when wind-splash may move the sediment a matter of cm (Warburton, 2003), and can be integrated into water erosion, however this is likely untrue for summer when peat desiccation occurs and small peat particles could be subject to longer-range aeolian transport or in farmed fens that may be extensively ploughed (Cumming, 2018). This is one of the many uncertainties and context dependencies that needs to be addressed to better partition fluxes of CO<sub>2</sub> against other apparent losses of peat from eroded systems (Fig.2).

Figure 2: Erosion gully with visible drying of the bare peat and rill erosion/water flow carrying POC downstream, vertical arrow indicates direction of peat surface change. Right- potential co-occurring forces that result in measurable surface retreat of bare peat.

#### 2.2 Respiration and loss to CO<sub>2</sub>

Peat is formed from partially-degraded remains of plant material that have remained in situ from centuries to millennia, as a result of a shallow water table. Once this relatively undegraded material (Biester et al., 2014) is released from hydrological controls on its decomposition -i.e. it is exposed to an abundance of oxygen, new microbial communities and warmer temperatures- it has potential to rapidly degrade (Artz et al., 2008; Robinson et al., 2023). Coupled with an absence of primary productivity on the unvegetated peat, exposure of peat to aerobic conditions can lead to a sizable CO<sub>2</sub> flux (but a reduced CH<sub>4</sub> flux compared to vegetated bog) (Artz et al., 2022; Evans et al., 2021b). The CO<sub>2</sub> flux from bare peat is termed "wastage" (Evans et al., 2006) and is difficult to measure from eroding peat surfaces because they are highly dynamic and remote, therefore wastage is often quantified through process of elimination after independently-measured fluxes of wind and water erosion are subtracted from measured peat loss (Evans et al., 2006; Francis, 1990). Francis (1990) observed greatest surface retreat rates (SRR) of bare peat in the summer when peat temperatures were highest, despite the highest flux of eroded sediment in the winter, leading to the conclusion that direct CO2 flux from the bare peat was a major loss pathway. Among the few papers that have estimated peat wastage, there is large uncertainty as to the importance of this process, with estimates ranging from around 56-81% of the measured SRR (Francis, 1990), to 5.75% measured directly through mass loss (Pawson, 2008). Evans et al. (2006) estimated wastage rates between 30-46 % depending on site characteristics (calculated by subtraction (Evans et al., 2006)). Chamber based CO<sub>2</sub> flux measurements estimate bare peat in gullies to be a small summer source of CO2, although the literature is sparse with very few year-round measurements and/or modelling studies to estimate annual budgets (Dixon et al., 2014; Clay et al., 2012; Gatis et al., 2019). However, at one blanket bog site, ecosystem respiration in vegetated gullies was found to be the highest within the landscape (Mcnamara et al., 2008), reinforcing how variable this flux could be.

#### 2.3 Subsidence

Erosion gullies cause drainage of the bare and surrounding vegetated peat (Daniels et al., 2008). Low water-table depths can cause subsidence of peat through reduction of water in peat soil pore spaces and the associated reduction of buoyancy. As a result, pore spaces collapse, the density of peat increases, and the peat loses elevation. This phenomenon is well understood in peatlands where drainage is implemented for plantations in the tropics (Hooijer et al., 2012), agriculture in temperate systems (Hutchinson, 1980) or on forested or drained temperate-boreal bogs (Defrenne et al., 2023; Sloan et al., 2019; Williamson et al., 2017). When drainage is imposed on bogs they can subside at a range of rates, from 0.5m over 100 years (Defrenne et al., 2023) to 2m in 20 years (Hooijer et al., 2012). In the Flow Country of Scotland, drainage for forestry caused a 57 cm average subsidence on a blanket bog over 50 years (Sloan et al., 2019), which could have been caused by a combination of heavy machinery (required for forestry) and the biotic drivers of peat carbon loss caused by trees (Defrenne et al., 2023) in addition to subsidence. Bare peat within eroding blanket bogs have SRR that are double (see next section and Li et al. (2018b)) the rate of elevational change attributed to subsidence in drained, forested systems (Sloan et al., 2019), and this elevation change occurs in the absence of factors associated with forestry. Nonetheless, subsidence of peat could be an important factor for apparent surface retreat from eroding peatland systems, and by not accounting for it we may overestimate carbon losses as direct CO<sub>2</sub> or POC into the fluvial system.

To try to account for this potential covariation with CO<sub>2</sub> loss, subsidence and POC losses it could be informative to compare SRR at multiple points within a peatland system. After drainage, subsidence and CO<sub>2</sub> loss could be a considered a smooth linear or nonlinear process (Hooijer et al., 2012), consistently measurable across relatively large areas of drained land. Therefore, if subsidence was the driving factor behind bare peat SRR one would expect an even SRR across all spatially distributed observation points (depending on topography and drainage). In reality, SRR is stochastic and spatially variable within very small areas (< 1 m<sup>-2</sup>). Eroding blanket bogs are often on slopes and therefore subject to strong lateral surface flow of water (Evans and Warburton, 2008). Therefore, it is likely that water erosion is the primary cause of SRR in blanket bog systems. Methods for measuring peatland subsidence at high resolution are now available at low cost (Evans et al., 2021a), hence the relative role of subsidence of the whole peat system could be compared with loss rates at specific points on the bare peat, as traditionally measured by erosion pins. These scalable metrics for peat erosion/wasting and subsidence should be prioritised to help better estimate direct CO<sub>2</sub> emissions from bare peat. The drivers of SRR potentially include direct oxidation by microbial communities in aerobic conditions, subsidence resulting from reduced peat buoyancy and erosion (Fig. 2), the relative proportions of which will influence how much CO<sub>2</sub> emission can be attributed to SRR.

## 2.4 Estimating CO<sub>2</sub> Emissions from Bare Peat

To evaluate potential direct CO<sub>2</sub> flux to the atmosphere from bare peat surfaces (termed 'wastage' (Evans et al., 2006)), we assumed no subsidence (while acknowledging this may cause overestimates of other losses) and applied emissions factors to SRR data compiled by Li et al. (2018b). We calculated a median SRR of 18.9 mm yr<sup>-1</sup> for UK eroding blanket bogs from 22 datasets that contributed to the review by Li et al. (2018b) (Table 1). We then applied a best estimate of 35 % wastage rate (Evans et al., 2006), although this could vary between 5% (Pawson, 2008) and 80% (Francis, 1990), and UK average peat bulk density of 0.13 g cm<sup>-3</sup> for peat soils between 30-100 cm and carbon content of 53% (extracted from UK soil Database (Frogbrook et al., 2009)) to estimate CO<sub>2</sub> loss from bare peat surfaces of 16.7 tCO<sub>2</sub> ha<sup>-1</sup> yr<sup>-1</sup>, assuming that all gaseous carbon losses from these exposed surfaces is CO<sub>2</sub> (Table 1).

We scaled the CO<sub>2</sub> flux per area bare peat to the catchment scale by assuming 15 % bare peat area combined with 85% of the catchment is 'Modified bog' which covers typical heather-dominated bogs and which currently carries an average CO<sub>2</sub> emission factor of 0.03 t CO<sub>2</sub> ha<sup>-1</sup> yr<sup>-1</sup> (Evans et al., 2022) The assumption of 15 % bare peat in eroding blanket bogs is based on the UK average bare peat cover in these systems (Evans et al., 2017). The composite CO<sub>2</sub> flux for the landscape from our estimate from bare peat (15% at 16.7 tCO<sub>2</sub> ha<sup>-1</sup> yr<sup>-1</sup>) and average net ecosystem exchange estimates for vegetated 'modified bog' (85% at 0.03 tCO<sub>2</sub> ha<sup>-1</sup> yr<sup>-1</sup>) results in an estimate of 2.5 tCO<sub>2</sub> ha<sup>-1</sup> yr<sup>-1</sup> for the landscape. This represents a potentially large flux of CO<sub>2</sub> from peat bogs to the atmosphere. Although these calculations are based on very limited data, this rough estimate is comparable to a recently published paper where authors measured net ecosystem exchange of 3.6 tCO<sub>2</sub> ha<sup>-1</sup> yr<sup>-1</sup> over an eroding blanket bog with approximately 15 % bare peat cover (Artz et al., 2022). Similarly, a former peat extraction site in Quebec with low vegetation coverage represented a large carbon source of between 5.8 and 8.7 t CO<sub>2</sub> ha<sup>-1</sup> yr<sup>-1</sup> (Rankin et al., 2018), indicating that bare peat could be a large direct source of CO<sub>2</sub>.

To our knowledge, only a single study has measured oxidation of bare peat in the field, via mass loss (Pawson 2008) (although see Bell et al. (2018) for an example of short term peat decomposition under controlled conditions). In the same way that litter bags (and to a lesser extent humus bags (Adamczyk et al., 2019)) have been deployed across the worlds' ecosystems to estimate litter decay rate (Zhang et al., 2008), the peatland community could make a concerted effort to directly measure peat mass loss to address this evidence gap. This cheap and scalable approach could help inform us of an important flux of carbon from the system. Alternatively, or in parallel, an expanded deployment of combined SRR and downstream sediment flux could infer wastage rates and expand this sparse dataset (Francis, 1990). While acknowledging the inherent topographical complexity of eroding peatland landscapes introducing additional uncertainty to eddy covariance measurement (Artz et al.2022), we recommend additional large-scale monitoring of CO<sub>2</sub> fluxes from eroding bogs. Currently, there is only one available 'flux tower' observational data set from eroding peatland systems (Cairngorm Mountains, Scotland (Artz et al., 2022). Additional

monitoring could therefore reduce uncertainty regarding the importance of bare peat for direct CO<sub>2</sub> fluxes in other climates, bare peat extents and management scenarios.

Table 1: Measured Surface retreat rate (SRR) and estimated direct CO<sub>2</sub> and POC losses from bare peat. Catchment scale net ecosystem exchange (NEE) of CO<sub>2</sub> and POC losses for an eroding bog based on an assumption of 15% bare peat cover compared to measured CO<sub>2</sub> NEE (measured by Eddy Covariance (Artz et al., 2022) and POC losses (measured by sediment loss (Li et al., 2018b)) at catchment scales.

|                     | SRR                 | Direct CO <sub>2</sub> flux per area bare peat     | POC flux per area<br>bare peat | CO₂NEE Flux catchment scale | POC Flux catchment scale |
|---------------------|---------------------|----------------------------------------------------|--------------------------------|-----------------------------|--------------------------|
| Units               | mm yr <sup>-1</sup> | tCO <sub>2</sub> ha <sup>-1</sup> yr <sup>-1</sup> |                                |                             |                          |
| Estimated flux from |                     |                                                    |                                |                             |                          |
| erosion rates       | 18.9                | 16.7                                               | 31.1                           | 4.6                         | 4.7                      |
| Measured fluxes     |                     |                                                    |                                |                             |                          |
| (Artz et al., 2022) |                     |                                                    |                                | 3.6                         |                          |
| Measured fluxes (Li |                     |                                                    |                                |                             |                          |
| et al., 2018b)      |                     |                                                    |                                |                             | 0.7                      |

# 3 When peat erodes, how much is resedimented, oxidised or continues to be transported out of the erosion complex?

Erosion systems in peatlands are complex, often starting with narrow 'V' shaped gullies which over time become 'U' or trapezoidal- shaped gullies as lateral erosion predominates once erosion reaches the base of the peat (Evans and Warburton, 2008), with eroded peat moving through this system on its way to the headwater streams (Fig. 3). Based on sediment flux data compiled by Li et al. (2018b), we calculated 0.7 tCO<sub>2</sub>e ha<sup>-1</sup> yr<sup>-1</sup> loss of POC at the outlet of eroding blanket bogs (by applying a 53 % conversion factor for carbon content and a median sediment yield of 35 t km<sup>-2</sup> yr<sup>-1</sup>). This rate is over six times lower than the POC flux that would be lost from peatlands based on our SRR rates estimates (Table 1). Due to scale-dependency of processes (see Li et al. (2018b) for a detailed discussion on this), direct comparison such as this should be cautioned against. Nevertheless, this difference raises important questions about how peat carbon is processed within the peatland system with potential important implications for carbon budgets in eroding peatland systems. Peat can transit through wider 'U-shaped' gullies in a matter of hours; however, a proportion is likely to resediment within the system for years where it will be further oxidised by the microbial community. How long peat sediment is retained in the system and the conditions it is exposed to will determine its decomposition rates, how much moves on out of the system and how much is retained for the long term (Fig.3).

Repeated drone-based remote sensing approaches are highlighting the amount of redeposition of peat sediment within peatlands. These approaches not only measure the geomorphic loss of peat on gully walls but also the vertical accumulation of peat in wider gullies downstream (Li et al., 2019; Glendell et al., 2017). One of the next important questions to answer is what is the mean residence time of peat particles within the system and how much decomposition occurs while the sediment is in this new environment? We know that areas that contain wide gullies are strong sources of CO<sub>2</sub> (Artz et al., 2022), as are areas of bare peat in former peat extraction sites (Rankin et al., 2018). However, bare gullies were measured to be only marginal sources of CO<sub>2</sub> with low metabolic activity in general (Dixon et al., 2014; Gatis et al., 2019). Using eddy covariance at high temporal resolution, conversely, Artz et al. (2022) observed peaks of ecosystem respiration at their eroded sites, suggesting that there are abiotic triggers that cause a pulse of CO<sub>2</sub> flux from the peat. Therefore, it is important to collect more data across a range of bare peat systems and conditions to understand the likely loss rates via CO<sub>2</sub> in these gully complexes.

Vegetation cover on the gully floor is a key control on the rate of resedimentation, as higher cover of vascular plants and bryophytes will hold up the flow of water and suspended sediment (Crowe et al., 2008; Harris and Baird, 2019; Milner et al., 2021), resulting in a reduced POC yield at the outflow (Evans et al., 2006). Erosion systems often reach a revegetated-base state with low POC flux rates at the outlet (Evans et al., 2006), however, it is not clear whether the trapped peat sediment is lost as CO<sub>2</sub> higher up in the peatland system. The turnover of resedimented material may depend on the vegetation cover of the gully floor, and its potential to trap freshly produced POC. Vegetated gullies were found to be 'hotspots' for ecosystem respiration in an eroding blanket bog (Mcnamara et al., 2008). Another naturally revegetated site had almost neutral CO<sub>2</sub> exchange as a result of high primary productivity being balanced by ecosystem respiration rates that were almost four times higher than in a bare gully system (Dixon et al., 2014). For both examples much of the ecosystem respiration flux will be from plant respiration, linked to vigorous growth associated with revegetated gully bases, however a significant amount could be from the resedimented peat. We understand that where peat is deposited on downstream floodplains (Alderson et al., 2019), it is turned over rapidly in aerobic conditions (Alderson et al., 2024) and up to 80% of POC deposited on a floodplain downstream of a peatland catchment will be mineralised to CO<sub>2</sub> within 30 years (Evans et al., 2013). Similar emission rates could be occurring in revegetated gully floors upstream of these floodplains. Microbial activity is likely to be lower in the gullies than downstream floodplains due to higher altitude (and therefore lower temperature), higher moisture (often saturating and anoxic) and potentially different plant communities and associated microbial communities. However, CO<sub>2</sub> flux from redeposited POC within peatlands could still be substantial and this knowledge gap could be resolved by measuring the isotopic signature of the CO<sub>2</sub> produced to partition into autotrophic and heterotrophic sources.

Figure 3: A schematic showing the transit of POC through multiple downstream systems with the pathways that carbon can take at different stages: converted to CO<sub>2</sub> (red), deposited in a long-term store (green) and transited to the next system as POC (blue). Percentage fluxes at each stage represent (a) best estimate of 35 % of SRR as 'wasting' CO<sub>2</sub> flux from bare peat surfaces (Evans et al., 2006) and (c) 70 %, the post-peatland export POC- CO<sub>2</sub> conversion factor determined by (Evans et al., 2013).

## 4 Concluding comments

Depending on the extent of bare peat within a peatland, and the local slope and wind conditions, erosion can be the dominant pathway for carbon loss (Evans et al., 2006). Peat that is lost through erosion has potential to be degraded to CO<sub>2</sub> at various stages on its transit as POC. Due to the complex biophysical processes and interactions that cascade from peat erosion there is very high uncertainty around the emissions that occur as a result. There is a risk of both under- and overestimating emissions from peat erosion, depending on the characteristics of the site. These uncertainties feed through to uncertainties in national peatland emissions reporting (Ipcc, 2014) and estimates of emissions reductions that can be achieved through bare peat restoration and revegetation. There are mismatches in data on sediment/carbon fluxes at certain points and across scales on the journey of peat after it is eroded. These mismatches can be addressed by applying scalable measurements at key junctures in the peat sediment's transition out of the peatland. These measurements should be biologically and biogeochemically focussed on the processes that mineralise peat sediment to CO<sub>2</sub>, and ultimately cause the emissions that we need to quantify. Climate change is driving increasingly severe erosive forces across all peatlands, from increased storminess to decreasing permafrost stability. Therefore, emissions arising from peat erosion are likely to have an increasingly important role in the carbon balance of all peatlands.

275

265

270

#### **Author Contribution**

Conceptualisation: TP & RA. Writing – review & editing: All authors.

#### **Competing interests**

The authors declare that they have no conflict of interest

# 280 Financial Support

Rural and Environment Science and Analytical Services Division Grant agreement or award number: JHI-D3-2

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
