# Peer review of "Ideas and Perspectives: Potentially Large but Highly Uncertain Carbon Dioxide Emissions Resulting from Peat Erosion"

_EGUsphere, 2025_

## Author Comment (AC1)

Response to Reviewer 1

(Reviewer comments in blue, our response in bold black and quoted text in black italics)

This paper highlights the importance, but also the lack of knowledge around particulate organic carbon erosion from peatlands and the contribution this could make to CO2 emissions as these degrade. This is an interesting paper which will be of interest to a broad audience.

I am left wondering how the DOC pathway fits into this model of C loss and the relative importance of wasting, DOC and POC for C loss. Some discussion of how these are connected and an acknowledgement that POC is not the only fluvial C export would be helpful.

**We agree that DOC is an important carbon loss pathway from peatlands and have now acknowledged this in the first paragraph as follows:**

'*Peatlands are important sources of fluvial carbon including particulate organic carbon (POC), dissolved organic carbon (DOC) and dissolved gases (Billett et al., 2015; Rosset et al., 2022). Previous studies have suggested that the relative roles of these fluvial forms are typically ~15-40% of $CO_2$ equivalent net ecosystem exchange (Dinsmore et al., 2010; Roulet et al., 2007; Billett et al., 2010). However, POC flux is particularly high from peatlands where vegetation cover is partial (Evans et al., 2006) and in these systems POC can contribute > 80 % of the fluvial flux (Pawson et al., 2008) while a lack of vegetation will also be associated with a reduced terrestrial C uptake across the peatland and potentially to enhanced direct losses to the atmosphere. Given such large potential contributions to C losses, it is critical that more studies acknowledge the POC pathway in the carbon budget. Previous studies have suggested that both DOC and POC are metabolised to $CO_2$ in the fluvial system to some degree, with current best estimates between 50 – 90% conversion for POC and 80 -100% for DOC (Evans et al., 2013).* However, most studies focus on terrestrial gas fluxes or aquatic DOC fluxes. Hence, the various pathways for POC storage, transport or transformation to $CO_2$ are not well studied (Palmer et al., 2016).

L29 I would remove particularly as this makes it seem a UK focused issue which is then contradicted by the paragraph starting l45

**We agree, we have replaced this text with the text above to comment more broadly on carbon losses from peatlands**

L55/56 a reference for the calculation of emissions from POC should be included here

**In response to RC2 on making our paper more generally applicable to all peatlands that are eroding, we included the IPCC 2013 Wetlands supplement equation on emissions from POC**

L121 typo but -> by

**We will apply this correction**

L158 https://www.sciencedirect.com/science/article/pii/S0016706117317275 measures mass loss from litter bags in a UK peatland

**We thank the reviewer for this reference and now refer to it, however, we note that this paper was a 60 day lab-based incubation of peat that will not undergo the variable temperature, moisture and physical disturbance that peat would during a multiannual study of decomposition in the field.**

Table 1 – the title is very long and repeats much of the text in paragraph starting line 144, I would suggest putting more detail in the main text and shortening the table caption.  If you wish to highlight this calculation, then perhaps convert it into a workflow figure.

**We have integrated this long figure legend into the text around line 144 as follows:**

*'To evaluate potential direct $CO_2$ flux to the atmosphere from bare peat surfaces (termed 'wastage' (Evans et al., 2006)), we assumed no subsidence (while acknowledging this may cause overestimates of other losses) and applied emissions factors to SRR data compiled by Li et al. (2018). We calculated a median SRR of 18.9 mm $yr^{-1}$ for UK eroding blanket bogs from 22 datasets that contributed to the review by Li et al. (2018) (Table 1). We then applied a best estimate of  35 % wastage rate (Evans et al., 2006), although this  could vary between 5% (Pawson, 2008) and 80% (Francis, 1990), and UK average peat bulk density of  0.13 g $cm^{-3}$ for peat soils between 30-100 cm and carbon content of 53% (extracted from UK soil Database (Frogbrook et al., 2009)) to estimate $CO_2$ loss from bare peat surfaces of 16.7 $tCO_2$ $ha^{-1}$ $yr^{-1}$, assuming that all gaseous carbon losses from these exposed surfaces is $CO_2$ (Table 1).*

*We scaled the $CO_2$ flux per area bare peat to the catchment scale by assuming 15 % bare peat area combined with 85% of the catchment is 'Modified bog' which covers typical heather-dominated bogs and which currently carries an average $CO_2$  emission factor of 0.03 t $CO_2$ $ha^{-1}$ $yr^{-1}$ (Evans et al., 2022) The assumption of 15 % bare peat in eroding blanket bogs is based on the UK average bare peat cover in these systems (Evans et al., 2017). The composite $CO_2$ flux for the landscape from our estimate from bare peat (15% at 16.7 $tCO_2$ $ha^{-1}$ $yr^{-1}$) and average net ecosystem exchange estimates for vegetated 'modified bog' (85% at 0.03 $tCO_2$e $ha^{-1}$ $yr^{-1}$) results in an estimate of 2.5 $tCO_2$ $ha^{-1}$ $yr^{-1}$for the landscape. This represents a potentially large flux of $CO_2$ from peat bogs to the atmosphere. Although  these  calculations are based on very limited data, this rough estimate is comparable to a recently published paper where authors measured net ecosystem exchange of 3.6 $tCO_2$ $ha^{-1}$ $yr^{-1}$ over an eroding blanket bog with approximately 15 % bare peat cover (Artz et al., 2022). Similarly, a former peat extraction site in Quebec with low vegetation coverage represented a large carbon source of between 5.8 and 8.7 t $CO_2$ $ha^{-1}$ $yr^{-1}$ (Rankin et al., 2018), indicating that bare peat could be a large direct source of $CO_2$.'*

**And shortened the table legend to read as follows:**

*'Table 1: Measured Surface retreat rate (SRR) and estimated direct $CO_2$ and POC losses from bare peat. Catchment scale net ecosystem exchange (NEE) of $CO_2$ and POC losses for an eroding bog based on an assumption of 15% bare peat cover compared to measured $CO_2$ NEE*

*(measured by Eddy Covariance (Artz et al., 2022) and POC losses (measured by sediment loss (Li et al., 2018)) at catchment scales.'*

**The details of how POC fluxes are estimated are still outlined in section 3 at Line 190**

Concluding remarks – needs a statement between the two sentences linking POC erosion to CO2 emissions.

**We agree with the reviewer and have added the text so the sentences are connected as follows:**

[revised manuscript text omitted]

---

## Author Comment (AC2)

Response to Reviewer 2

(Reviewer comments in blue, our response in bold black and quoted text in black italics)

Generally the article is well written and does an effective job of synthesising research in an area of interest in peatland carbon cycling. I find the explanations to be clear and the estimations to be revealing and valid.

The problem for me is that it is unclear if the focus of the article is UK blanket bogs or peatlands globally. I initially read the introduction as being partly a call for more POC-erosion emission research internationally to match that in the UK. However, by the end of the article I was left unsure if the authors were interested outside of a blanket bog setting. I understand it is necessary to draw mostly on research from the UK where this has been a greater focus. Yet if a global outlook is part of the purpose of this article then some attempt needs to be made to relate those findings to other peatland types found globally, and this needs to be done consistently throughout the article not just acknowledged somewhere. The non-UK erosion examples provided are interesting but I would appreciate more conjecture from the authors on the prevalence, type and importance of erosion and POC transport for emissions in different biomes relating to their typical peatland types and topography.

**This is a very important point. Our aim was to think about peatland erosion and POC more broadly but we agree that the focus on processing of POC in the latter half of the paper gives a impression that this is a UK-only phenomenon. We have rectified this issue by outlining the IPCC 2013 Wetlands supplement equation that was designed to be applicable in any eroding peatland and that the large sources of uncertainty are the flux of POC from bare peat and the conversion factor of POC to $CO_2$ in more generic terms as follows:**

*'The 2013 IPCC wetlands supplement (Ipcc, 2014) present a general calculation for a POC emissions factor ($EF_{POC}$) for all peatlands and drained organic soils. This generic model, although primarily based on evidence from the UK, was designed for any peatland soil that had suffered significant disturbance that lead to bare peat, including drainage, burning, peat extraction and conversion to arable land as follows (Ipcc, 2014):*

$$EF_{POC} = POC_{FLUX\ BAREPEAT} \times PEAT_{BARE} \times Frac_{POC-CO_2}$$

*Where: $POC_{FLUX\ BAREPEAT}$ is the POC flux per area of bare peat surface, $PEAT_{BARE}$ is the area of bare peat and $Frac_{POC-CO_2}$ is the conversion of POC to $CO_2$ following export from the peatland.*

*Mapping of bare peat extent at high resolution is progressing (Macfarlane et al., 2024) but the underpinning data for estimating emissions associated with bare peat are highly uncertain (Evans et al., 2013) as the flux depends on specific fluvial mixing events in time and space (Palmer et al., 2016). We argue that this calculation has two major sources of uncertainty which are critical to resolve to confidently quantify emissions that arise from peat erosion. Firstly, the flux of POC from bare peat at the source is only one part of peat volume loss - quantification of the relative contribution of direct $CO_2$ loss, subsidence and erosion to surface retreat rates will give rise to better quantification POC loss via erosion. Eroded peat will potentially be processed and mineralised in multiple environments, from headwater streams, floodplains to rivers and the*

*ocean (Evans et al., 2013; Zhou et al., 2021). Therefore, the second source of uncertainty is the fraction of eroded peat/POC that is converted to $CO_2$. This can be addressed by considering the environments and organisms that interact with it while in transit over various timescales.'*

**We also now include the following text at line 37 to acknowledge that the majority of peat erosion work has been carried out in the UK, to point out other regions where peat erosion is significant and that all areas will likely suffer worse erosion as climate change continues to impact extreme weather patterns:**

*'In the past century, most of the peat erosion and post-erosion POC research has been conducted in the UK. However, peat erosion is a pressing or emerging problem for peatland systems around the world, with potential for massive carbon losses and climate feedbacks (Fig.1). Potential erosion hotspots are occurring in different environmental and management contexts around the world from drained forestry sites which may have relatively low areas of exposed peat (Marttila and Klove, 2010) to industrial extraction sites with almost complete bare peat cover (Campbell et al., 2002). At the extreme end of erosion, collapse of inland permafrost systems in the arctic and boreal regions (Swindles et al., 2015) can cause localised rapid erosion and movement of soil carbon via thaw slumps (Lamoureux et al., 2014; Pizano et al., 2014), with potential for high emissions as the mobilised carbon becomes available to decomposer organisms in freshwater environments (Li et al., 2024). In contrast, arctic permafrost coastal erosion and coastal-adjacent thaw slumps, which are occurring at an alarming rate in response to rapid warming around the Arctic Ocean, are depositing carbon directly into the ocean (Lantuit and Pollard, 2008; Lantuit et al., 2012). Equally, In the tropics of Asia, coastal erosion of peatlands is causing large direct fluxes of peat to the ocean (Kagawa et al., 2024), but there are also examples of inland peat erosion in Asia which will generate POC that is primary processed in terrestrial systems (Wang et al., 2019).*

*Peat erosion is clearly progressing in a variety of contexts and at different rates, but in every case it will be exacerbated by climate change and associated extreme weather events (Zhao et al., 2024). This is why IPCC reporting of emissions needs to move towards a more nuanced understanding of POC turnover than the broad downstream POC-$CO_2$ conversion rate of 70% (based on UK examples (Ipcc, 2014)). Depending on the context, biome and global location, estimated emissions resulting from peat erosion could vary significantly from currently reported rates.'*

**Furthermore, we have edited figure 3 to be more generic and applicable to any peatland erosion scenario because we appreciate the previous version was too specific to erosion of UK blanket bogs:**

[Figure]

We have also added the sentence below to the concluding paragraph to make the point that climate change threatens increased erosion rates for all peatlands:

*'Climate change is driving increasingly severe erosive forces across all peatlands, from increased storminess to decreasing permafrost stability. Therefore emissions arising from peat erosion are likely to have an increasingly important role in the carbon balance of all peatlands.'*

In a similar fashion, the title uses "greenhouse gas emissions" but a text search finds exactly one mention of CH4 in parenthesis and none of N2O, as such I suggest the title is changed to $CO_2$.

**We agree that the processes we outline are relevant to $CO_2$ fluxes. We have edited mentioned references to 'greenhouse gas emissions' to '$CO_2$ emissions' or 'fluxes' where appropriate. We also propose editing the title to 'Carbon Dioxide Emissions' to more precisely encapsulate the issues we discuss.**

On reviewing the emissions factors associated with different peatland conditions and the compound $CO_2$ emissions from bare peat and vegetated 'modified bog' we noticed an error where we had listed emissions as 2.51 t $CO_2$e ha$^{-1}$ yr$^{-1}$ . In reality this figure is for all carbon loss pathways, including DOC and $CH_4$ (Evans et al. 2022). We now *use 0.03 tCO$_2$ ha$^{-1}$ yr$^{-1}$* for the $CO_2$ component of emissions from modified bog which reduces our estimate of 4.6  t $CO_2$ ha$^{-1}$ yr$^{-1}$ to 2.5 t $CO_2$ ha$^{-1}$ yr$^{-1}$ but the point still stands that our estimate is similar to measured rates (3.6t $CO_2$ ha$^{-1}$ yr$^{-1}$ by eddy covariance (Artz et al. 2022) (see the text in response to RC1 for the full context and calculations).

I also think more discussion is needed on the role of human land use pressures in influencing peatland erosion. I have understood this may be relevant in UK blanket bog erosion. It is certainly relevant for peatland systems that do not naturally exhibit significant erosion.

**We have added text to the second paragraph (line 37) that reads as follows:**

'*The onset of peatland erosion can be traced back over a thousand years (Evans and Warburton, 2011). It is hypothesised that there is a 'threshold process' whereby the peat changes from a stable, intact state to an unstable, erosional state. Some propose that erosion is a natural termination after thousands of years of peat accumulation resulting in instability of the peat mass (Conway, 1954; Pearsall, 1956; Colhoun et al., 1965). Others argue that much of the erosion has resulted from anthropogenic pressures, including burning (Yallop et al., 2009), overgrazing (Wilson et al., 1993),  artificial drainage installation (Worrall and Evans, 2009; Holden et al., 2007), and atmospheric pollution (Yeloff et al., 2006).*

In conclusion, there is a mismatch between the articles implied focus and it's actual focus, as such one or the other needs to be altered. I suggest the authors attempt to relate the different aspects of emissions from peatland erosion discussed to other peatland types and settings internationally throughout the article. Or, if I have misunderstood and the article was always supposed to be blanket bog focussed, then I suggest that they edit the abstract, introduction and title to make this clearer.

**We agree with the sentiment of the comment. As detailed above we have edited the language and figures and included detailed references to the IPCC guidance on $CO_2$ emissions from POC loss to be more generically applicable to all eroding peatlands. While much of the work on the subject of POC processing post initial erosion has been conducted in the UK, we now refer to  Li et al. (2024) on remote sensing of retrogressive thaw slumps in permafrost regions both in highlighting progress in monitoring (Line 66) and with regards to early thoughts on  post-erosion carbon cycling (Line 186). However, as noted by the reviewers, we are trying to strike a balance between the volume of understanding that has been generated in the UK and the general applicability of these ideas to other eroding peat systems.**

Line 85 "maybe" should be "may be"

**We will apply this change**

References

[revised manuscript text omitted]